# Nano Freezing–Thawing of Atlantic Salmon Fillets: Impact on Thermodynamic and Quality Characteristics

**DOI:** 10.3390/foods12152887

**Published:** 2023-07-29

**Authors:** Wenxuan Wang, Wenzheng Li, Ying Bu, Xuepeng Li, Wenhui Zhu

**Affiliations:** College of Food Science and Engineering, Bohai University, No. 19, Keji Road, Jinzhou 121013, China; wwx2987542477@126.com (W.W.); liwenzheng1113@163.com (W.L.); buying130@126.com (Y.B.); xuepengli8234@163.com (X.L.)

**Keywords:** salmon fillets, magnetic nanoparticles (MNPs), freezing–thawing methods, thermal characteristics, protein oxidation

## Abstract

The presence of magnetic nanoparticles (MNPs) suppresses ice nucleation and growth during freezing and thawing. In this study, the effects of MNPs-assisted cryogenic freezing integrated with MNP-combined microwave thawing (NNMT) on the thermodynamic and quality changes of salmon fillets were investigated. Results have shown that NNMT raises *T_g_* (glass transition temperature) and *T_max_* (transition temperature), thus improving the storage stability of salmon fillets. MNPs-assisted freezing and thawing treatment, especially NNMT treatment, significantly improved the water holding capacity, texture, color, and other quality characteristics of salmon fillets. In addition, the lipid and protein oxidation degrees of the NNMT treatment were the lowest, while the myofibrillar protein solubility of NNMT was the highest (87.28%). This study demonstrated that NNMT has minimal impact on the freezing–thawing quality of salmon fillets, making it a more suitable option for the preservation of aquatic foods.

## 1. Introduction

Atlantic salmon (*Salmo salar*) is a migratory fish widely distributed in the northern Atlantic Ocean and adjacent freshwater [1]. As the largest edible salmon species, Atlantic salmon can be cooked in plenty of ways, such as smoking, grilling, and sushi. High-quality Atlantic salmon is essential for processing and retail establishments, with shelf-life being a critical factor that significantly impacts the salmon processing industry [2]. The freezing of Atlantic salmon is a crucial step in ensuring its quality and safety in accordance with industrial protocols. Freezing is the most crucial method for storing aquatic products, as it effectively inhibits microbial growth and reproduction, prolongs shelf life, and serves as the primary form of importing/exporting marine products and facilitating inter-regional circulation. However, the physical and biochemical reactions during freezing lead to the structural destruction of seafood [3], especially ice crystal formation, which is the primary source of food damage caused by freezing [4]. Meanwhile, the subsequent thawing process will also cause drip loss and quality deterioration. Scholars traditionally focus solely on the study of freezing and thawing techniques, which may not be conducive to maintaining the quality of thawed aquatic products. Therefore, it is imperative to explore new technologies for both freezing and thawing processes.

Magnetic nanoparticles (MNPs) have been approved as a safe material by the U.S. Food and Drug Administration, and they have superparamagnetic properties, a large specific surface area, non-toxicity, and recyclability [5]. MNPs with a size of 50 nm have been reported to exhibit non-toxicity even after remaining in the body for a duration of four weeks at three different doses (100, 50, and 25 mg/kg) [6]. Fe_3_O_4_ MNPs have an excellent inhibitory effect on myofibrillar protein aggregation and oxidation when combined with microwave thawing [7,8]. Zhu et al. also demonstrated that nanowarming can reduce impairment to muscle fibers caused by ice crystals and reach the requirement of rapid and uniform thawing [9]. According to nucleation theory, the introduction of additional particles can alter the free energy barrier for nucleation, while nanomaterials have been shown to facilitate the low-temperature preservation of biological materials. Cai et al. utilized CS@Fe_3_O_4_ nanoparticles plus herring antifreeze protein (AFPs) solution to pre-treat fish meats before freezing and thawing, resulting in the effective inhibition of protein oxidation through the antifreeze properties of both CS@Fe_3_O_4_ nanoparticles and AFPs [10]. When the food is in a glassy state, the rate of deterioration reactions (recrystallization, protein oxidation, and lipid oxidation) decreases [11]. Wang et al. found Fe_3_O_4_ nanoparticles significantly promoted the rewarming of human stem cells and improved the cryopreservation outcome [12]. It is confirmed that MNPs could improve the vitrification efficiency of immature mouse oocytes as a vitrification aid [13]. Sharma et al. perfused a cryoprotective cocktail and silica-coated MNPs into a kidney and discovered that the growth of ice crystals could be avoided by nanowarming [14]. Hence, we propose the application of MNPs in the freezing–thawing process of aquatic products to enhance their glass transition temperature, inhibit ice nucleation and growth, and ensure superior quality.

In this paper, the impact of MNPs-assisted cryogenic freezing integrated with MNP-combined microwave thawing (nano freezing–thawing) on the thermal properties and quality attributes of salmon fillets was evaluated. The thermal characteristics of salmon fillets were studied by determining the freezing point, unfreezable water content, apparent specific heat, and protein transition temperature. The water holding capacity (WHC), color, texture profile analysis (TPA), oxidative denaturation of proteins, and lipid oxidation of salmon were determined to reflect the effect of nano freezing–thawing on salmon fillet quality.

## 2. Materials and Methods

### 2.1. Materials

Fe_3_O_4_ MNPs (20–50 nm, prepared by a hydrothermal method) were purchased from Aike Reagent (Sichuan, China). Atlantic salmon (*Salmo salar*) fillets were purchased from Shandong Meijia Group Co., Ltd. (Shandong, China). Each salmon fillet was cut into 7 cm × 7 cm × 3 cm pieces and weighed about 150 g. Subsequently, the salmon portions were put into polyethylene bags and laid in a −80 °C refrigerator for subsequent treatments.

### 2.2. Sample Preparation and Freezing–Thawing Treatment

Salmon fillets in frozen storage were randomly assigned to four groups for four freezing–thawing methods. The thawing process was considered complete when the central temperature of the sample reached 0 °C. The temperatures of the salmon fillets were monitored in real time by a temperature recorder (Applent Precision Instrument Co., Ltd., Changzhou, China), linked to a K-type thermocouple.

#### 2.2.1. FCST (Cryogenic Freezing Integrated with 4 °C Cold Storage Thawing)

The frozen salmon fillet was stored in a polyethylene bag, frozen at −80 °C for 72 h, then thawed in a refrigerator at 4 °C.

#### 2.2.2. FMT (Cryogenic Freezing Integrated with MNPs Combined Microwave Thawing)

The frozen salmon fillet was placed in a polyethylene bag and frozen at −80 °C for 72 h. The frozen salmon fillet was completely soaked in the MNPs solution (0.9 mg/mL). After that, a 500 W microwave oven (NN-DF392B, Panasonic, 350 × 299 × 199 mm, Osaka, Japan) was used to thaw the fillet. The parameters of the thawing process and the concentration of MNPs solution were selected according to Zhu et al. [9].

#### 2.2.3. NCST (MNPs Assisted Cryogenic Freezing Integrated with Cold Storage Thawing)

The frozen salmon fillet was completely soaked in the MNPs solution (0.9 mg/mL) at a mass ratio of 1:2 for 1 h at 4 °C, then samples soaked in MNPs were packed and placed at −80 °C for 72 h. The frozen salmon fillets were thawed in a 4 °C refrigerator.

#### 2.2.4. NNMT (MNPs Assisted Cryogenic Freezing Integrated with MNPs Combined Microwave Thawing)

The frozen salmon fillet was utterly soaked in MNPs solution (0.9 mg/mL) at a mass ratio of 1:2 for 1 h at 4 °C. Samples in the solution were removed and wrapped with polyethylene bags, then placed at −80 °C for 72 h. Subsequently, 72 h later, the frozen salmon fillet was completely immersed in the MNPs solution (0.9 mg/mL) and used a 500 W microwave oven to thaw.

### 2.3. Thermal Characteristic Parameters Determination

Briefly, 5–10 mg of freezing–thawing-treated samples was accurately weighed in a solid aluminum crucible. The freezing point, unfreezable water content, and apparent specific heat of the sample were determined according to Hamdami et al. by adopting a differential scanning calorimeter (DSC) (Q2000, TA Instruments, New Castle, DE, USA) [15].

#### 2.3.1. Freezing Point and Unfreezable Water Content

The freezing point appears on the heating curve. Due to the influence of thermal hysteresis, the phase transition peak temperature is determined as the freezing point. The unfreezable water content was determined and calculated as follows:(1)$$\alpha =\omega -\frac{∆HTf}{L}$$
where α—unfreezable water content, % (wet base); ω—water content, % (wet base); Tf—sample freezing temperature, °C; ∆HT*f*—freezing enthalpy of the sample at freezing temperature, J/g; and L—freezing enthalpy of pure water at freezing temperature, J/g.

#### 2.3.2. Apparent Specific Heat

The apparent specific heat of salmon fillets was determined by the three-line method. All heat flow signals were obtained from the DSC curves, and KCl was used as the reference material.

#### 2.3.3. Glass Transition Temperature (*T_g_*)

The *T_g_* of salmon fillets was measured by a double scanning program, which was slightly modified according to the scanning program of Shi et al. [16]. The specific procedures were as follows: The sample was cooled at 10 °C/min to −60 °C and equilibrated for 2 min. Then, the temperature was increased to 70 °C at 10 °C/min and equilibrated at 70 °C for 2 min. Afterward, the temperature was cooled to −60 °C at 10 °C/min and balanced at −60 °C for 2 min. Finally, the temperature was increased to 40 °C at the same rate. The DSC software was applied to analyze the heat flow curve and the numerical values of the initial glass transition temperature (*T_gi_*), midpoint glass transition temperature (*T_gm_*), and end glass transition temperature (*T_ge_*) were obtained. The *T_gm_* (I) value was determined as the glass transition temperature of the sample.

#### 2.3.4. Protein Thermal Stability

The thermal stability of salmon protein was determined using the DSC, and 5–10 mg of salmon meat was weighed in a solid aluminum crucible. The initial temperature of the sample was set at 25 °C, and the sample was heated to 80 °C at 5 °C/min. The TA analysis software was used to integrate the corresponding area to obtain the enthalpy (∆H) and transition temperature (*T_max_*) of the samples.

### 2.4. WHC

#### 2.4.1. Thawing Loss, Cooking Loss, and Centrifugation Loss

The thawing loss, cooking loss, and centrifugation loss were measured on the basis of Zhu et al. [9]. Briefly, the sample was cut into 2 cm^3^ pieces and weighed, then put into a cooking bag, heated in a water bath (85 °C, 20 min), weighed, and the cooking loss of the sample calculated. After freezing–thawing treatment, the samples were cut into 1 cm × 1 cm × 2 cm pieces and centrifuged (4 °C, 4000× *g*, 10 min). The quality of salmon samples before and after centrifugation was accurately weighed, and the centrifugation loss was calculated.

#### 2.4.2. Low-Field Nuclear Magnetic Resonance (LF-NMR)

The nuclear magnetic resonance relaxation time *T_2_* of salmon fillets was measured using an NMR Analyzer (NM2012, Shanghai), according to Zhu et al. [9]. The CPMG sequence was used to determine *T*_2_, with the specific parameters as follows: SFI (MHz): 22; P90 (μm): 16.5; SW (KHz): 200.

#### 2.4.3. Magnetic Relaxation Image (MRI)

Multiple spin-echo-sequence (MSE) pulses were used for 2D proton density imaging of salmon fillets. The specific parameters are FOV: 100; TR: 2000 ms; average: 4.

### 2.5. Color

The luminance value (*L**), redness value (*a**), and yellowness value (*b**) of salmon fillets were determined via a portable colorimeter (CR400, Konica Minolta, Tokyo, Japan).

### 2.6. Texture Profile Analysis (TPA) and Stress Relaxation Determination

The freezing–thawing salmon fillet samples were cut into 2 × 2 × 2 cm^3^ cube fish pieces, and the TPA mode of the texture analyzer (TA-XT PLUS, Stable Micro System, Britain, Godalming, UK) (test speed 1 mm/sec, compression ratio 30%, trigger force 5.0 g, cylindrical probe P50) was used to determine the texture characteristics. And the compression mode (test speed 1 mm/s, compression ratio 30%, trigger force 5.0 g, maintenance time 45 s, cylindrical probe P50) was used to determine the stress relaxation parameters of the sample.

### 2.7. Determination of Protein and Lipid Oxidation

#### 2.7.1. Myofibrillar Protein (MP) Extraction

Samples were minced and blended with four volumes of Tris-HCl solution (20 mmol/L, pH 7.2, *v*/*w*), then homogenized (6000 rpm, 30 s). The mixture was centrifuged using a centrifuge (THERMO, Thermo Company, Waltham, MA, USA) at 6500× *g* for 15 min. After centrifugation, the process was repeated with sediment as described above. The precipitate dissolved in four volumes of Tris-HCl-NaCl solution (20 mmol/L). Afterward, the compound was homogenized at 6000 rpm for 30 s and centrifuged at 6500× *g* for 15 min, and the supernatant was MP [17].

#### 2.7.2. Protein Oxidation

The total sulfhydryl content was quantified using the 5,5-dithio-bis (2-nitrobenzoic acid) (DTNB) method and calculated based on an extinction coefficient of 13,600 M^−1^ cm^−1^. The carbonyl content of proteins was determined by the 2,4-dinitrophenylhydrazine (DNPH) derivation method. The content of dityrosine was determined by the fluorescence spectrophotometer, and the corrected fluorescence values were obtained by dividing the measured fluorescence values by the protein concentration [9].

#### 2.7.3. Lipid Oxidation

The thiobarbituric acid (TBA) value was determined by the method of Zhu et al. [18]. Freeze-thawed salmon (5 g) was homogenized with 5 times the volume of a 5% (*w*/*v*) trichloroacetic acid (TCA) solution, and the supernatant was mixed with 0.02 mol/L TBA at a 1:1 (*v*/*v*), heated for 20 min. Finally, the absorbance of the solution at 532 nm was determined.

### 2.8. Protein Denaturation

#### 2.8.1. Particle Size

The concentration of MP was diluted to 1 mg/mL using the Tris-HCl-NaCl buffer, and the average particle size of the salmon MP solution was determined using a zeta potentiometer (90 Plus Zeta, Spring House, PA, USA) [9].

#### 2.8.2. Solubility of Protein

The protein solubility was calculated by the formula of a percentage of protein concentration before and after centrifugation [9].

#### 2.8.3. Sodium Dodecyl Sulfate-Polyacrylamide Gel Electrophoresis (SDS-PAGE)

The electrophoretic profile was analyzed using a discontinuous gel system consisting of a 12% separating gel and a 4% stacking gel, with SDS buffer in PAGE. After electrophoresis, the gel strips were stained with coomassie brilliant blue dye and subsequently decolorized until protein bands became clearly visible [9].

### 2.9. Statistical Analysis

All measurements in the experiment were repeated more than three times; the color, TPA, stress relaxation, and WHC measurements were repeated six times, and the results were presented as the mean ± standard deviation. SPSS 19.0 (SPSS Inc., Chicago, IL, USA) was used to perform a one-way analysis of variance (ANOVA) on the experimental data, and images were obtained through OriginPro 9.0.

## 3. Results and Discussion

### 3.1. Thermal Characteristic Analysis

#### 3.1.1. Freezing Point and Unfreezable Water Content

The unfreezable water content, freezing point, and freezing enthalpy of salmon fillets with different freezing–thawing methods are shown in Table 1. The unfreezable water content of salmon fillets increased from 12.68% (FCST) to 19.31% (NNMT). The increase in unfreezable water content reduces the freezable water content in the freezing process and prevents the growth of abundant ice crystals.

The peak temperature of phase transition is considered the freezing point temperature of salmon [15]. The freezing point of the aquatic product is associated with water content, soluble small molecule content, and tissue, mainly at −1.00~−2.20 °C. In this study, the freezing point of salmon fillets was −1.00~−1.35 °C. As the freezing point dropped, the corresponding enthalpy of the phase decreased, and it was propitious to pass through the zone of maximum ice crystal formation (−1.00~−5.00 °C). The freezing enthalpy of FCST was 194.55 J·g^−1^, and the freezing enthalpy of the NNMT sample decreased to 161.85 J·g^−1^. Hong et al. found that nanomaterials are more likely to reduce the freezing point of the solution compared with conventional ionic salts (e.g., NaCl) [19], and the freezing point of the NNMT sample was reduced by 0.35 °C, which may be due to the added nanoparticles’ excellent dispersion and the ability to promote heterogeneous nucleation to reduce the freezing point of the solution.

#### 3.1.2. Apparent Specific Heat (C_app_) and *T_g_*

As shown in Figure 1, the C_app_ of salmon fillets had no significant change with increasing temperature in the low-temperature zone (−40.00~−10.00 °C). Nonetheless, the C_app_ of salmon fillets rose significantly in the phase transition area (−10.00~5.00 °C), and the peak value appeared near the freezing point. The result was in agreement with the study of Hamdami et al. [15]. In addition, the peak value of the apparent specific heat of MNPs-assisted freezing of samples was generally lower than that of direct freezing of salmon fillet samples. It may be that the nanoparticles adsorb a large number of water molecules on their surface and form a film of water molecules around the nanoparticles, resulting in less free water and further influencing the latent heat of phase transition, which results in the descent of the C_app_ [20]. After the phase transition, the effect of temperature on latent heat and C_app_ diminishes when ice crystals completely transform into liquid water. In this study, the C_app_ of salmon fillets decreased and tended to be consistent in the four freezing–thawing methods.

Glass transition temperature (*T_g_*) is a second-order phase change temperature and one of the most significant parameters in the “glass transition” mechanism [21]. As shown in Figure 2, the temperature around −60.00 °C could be the *T_g_* of salmon protein, while the temperature around −20.00 °C was the *T_g_* of salmon lipid [22]. Compared with the FCST treatment group (Figure 2A), the *T_g_* of salmon lipids in the NCST, FMT, and NNMT treatment groups increased from −26.42 °C to −13.37 °C, −12.50 °C, and −12.20 °C, respectively. However, the changes in *T_g_* of salmon protein treated by MNPs-assisted freezing were not obvious. The moisture content in food significantly affects the *T_g_*, and the treatment of MNPs reduced the content of free water in salmon, which in turn increased the *T_g_* of salmon fillets. Many hydroxyl groups (–OH) are on the surface of Fe_3_O_4_ MNPs, and the greater the polarity, the greater the *T_g_* of the salmon [23]. And the increase in heating and cooling rates also increases *T_g._* According to the glass transition theory, the food is in a solid-like glassy state when the system temperature is below the *T_g_* [24], and all the deterioration reaction rates are decreased. Therefore, storage stability is best when the temperature is below *T_g_*. The optimal storage temperature of frozen food is T ≤ *T_g_*, yet most fish species have ultra-low *T_g_*. De La Cruz-Montoya et al. [25] found that the incorporation of magnetic nanoparticles can effectively enhance the *T_g_* of polymer nanocomposites. It has been demonstrated that the increase in *T_g_* observed in nanocomposites is a consequence of reduced polymer mobility, which results from both the fine dispersion of nanoparticles within the polymer matrix and the interfacial adhesion between the nanoparticle phase and polymer matrix [26]. However, several studies [27,28] have demonstrated that the incorporation of magnetic nanoparticles does not exert any influence on either the *T_g_* of polymers or the typical vitrification solution Vs55. Certain studies have demonstrated that an increase in non-freezable water content can lead to a reduction in the *T_g_*, which contradicts these experimental findings. In comparison to unfrozen water content, the influence of free water content on *T_g_* in this study may be more favorable for enhancing *T_g_* in salmon fillets. The specific reasons need to be further studied. This study confirmed that the treatment of NNMT can increase *T_g_* and enhance the storage stability of salmon fillets.

#### 3.1.3. Protein Thermal Stability

The *T_max_* parameter reflects the thermodynamic stability of proteins, while ∆H represents the energy required to disrupt intramolecular interactions during protein denaturation. Three characteristic absorption peaks can be observed in the DSC heat flow curve (Figure 3), which are related to the degeneration of myosin head (*T_max_*1), myosin tail (*T_max_*2), and actin (*T_max_*3). As shown in Table 2, the *T_max_*1 and *T_max_*3 of NNMT were significantly higher than FCST, FMT, and NCST (*p* < 0.05), while the *T_max_*2 of different treatment groups was not significantly higher (*p* > 0.05). There was no significant difference (*p* > 0.05) in the ∆H1 and ∆H3 among the FCST, FMT, NCST, and NNMT. Overall, the protein thermal stability of the NNMT treatment is the best. However, FCST showed the worst protein thermal stability, which may be caused by myosin subunit dissociation and actin oxidation.

### 3.2. Analysis of WHC

#### 3.2.1. Thawing Loss, Cooking Loss, and Centrifugation Loss

The increase in thawing loss is primarily attributed to the growth of ice crystals that compress cellular structures and muscle fibers, leading to water not being reabsorbed during the thawing process. As can be seen in Figure 4A_1_, the thawing loss of NCST was significantly lower than that of FCST, and that of NNMT was the lowest (9.27%) (*p* < 0.05). Heating induces myofibrillar protein degeneration, which destroys muscle fiber structure and causes an increase in cooking loss, including the loss of large amounts of moisture and small amounts of soluble material (protein, nucleotide, and free amino acid) in meat during heating. The cooking loss of NNMT and FMT groups was significantly lower than other treatment groups (*p* < 0.05), and there was no significant difference among FCST and NCST samples (Figure 4A_2_). During freezing, MNPs reduced the content of free water in salmon fillets, effectively inhibiting the formation of substantial ice crystals, which was not conducive to drip loss caused by ice crystals during thawing. The MNPs would shift to a paramagnetic state affected by microwaves; thus, the heat generated by the MNPs increased linearly, which accelerated the thawing rate of salmon fillets [18]. So compared with the FCST and NCST samples, the FMT and NNMT samples had relatively high WHC, while the NNMT group had the strongest WHC. The results of centrifugation and thawing loss are similar (Figure 4A_3_). The decrease in WHC is associated with myofibrillar damage and protein degeneration [29], and poor WHC initiates more severe protein oxidation [18].

The WHC of MNPs-assisted freezing and microwave thawing samples was better than that of samples not immersed in MNPs solution. In addition, the thawing loss, centrifugation loss, and cooking loss of the FCST sample were the highest. On the one hand, due to the long thawing time, the integrity of fish tissue is destroyed and the drip loss channel of meat is widened, resulting in the worst WHC. On the other hand, this trend is dependent on the phase transformation of the sample during freezing [30]. FCST cannot meet the requirements of rapid freezing when it passes slowly through the phase transition zone, causing large ice crystal formation and a reduction in WHC. However, NNMT treatment reduced the freezing point and promoted the formation of small ice crystals, which was conducive to improving the WHC of salmon.

#### 3.2.2. Water Distribution

The relaxation time *T*_2_ can reflect the mobility of water molecules in the sample. When water mobility in salmon is restricted, the *T*_2_ values decrease correspondingly [31]. As shown in Figure 4B, samples with different freezing–thawing methods showed different *T*_2_ values, indicating that water migration occurs in salmon. The hydrogen proton peaks represent three water phases, including bound water *T*_21_ (<10 ms), tightly bound to polar groups of macromolecules; immobilized water *T*_22_ (10~100 ms), which exists in the myofibrils grid; and free water *T*_23_ (100~1000 ms), which presents in the myofibrils lattice. The distribution and migration of various forms of water in salmon fillets under different freezing–thawing conditions can be reflected by the peak area percentage of water in different states.

In Figure 4C, P_21_, P_22_, and P_23_ represent the peak area proportions of *T*_21_, *T*_22_, and *T*_23_, respectively. Immobilized water is the most important form, accounting for about 95% of the total water content. In addition, free water *T*_23_ occupied about 2%, which could be lost easily under exogenic action, and excessive free water will adversely affect the preservation of aquatic products. At the same time, the proportion of bound water *T*_21_ is less than 2%, and the fluidity is the worst and the most stable. The proportion of immobilized water in the NNMT group was the largest, illustrating that the water retention of salmon fillets obtained by this freezing–thawing method was better. In the absence of an applied magnetic field, MNPs uniformly disperse and maintain their stability [32], promoting heterogeneous nucleation of the sample during freezing, generating free water backflow, and transitioning to immobilized water. The large percentage of free water in FCST showed that the extracellular ice crystals caused substantial non-reversible mechanical damage to the cell membrane, accelerating the immobilized water conversion into free water and leading to the reduction in WHC. It could also be attributed to the hydrolysis of structural proteins, resulting in the liberation of protein-bound water [33]. Furthermore, the quantity of immobilized water during cold storage thawing was found to be lower compared to microwave thawing, which is in line with the research conducted by Lan et al. [34]. In addition, free water provides convenience for microbial propagation and biochemical reactions, and too much free water is not conducive to food storage. The NNMT enhanced water–protein interactions, and salmon fillets treated with NNMT had the highest moisture content, which was in accord with the result of proton density images.

MRI has been popularly applied to study water distribution by visualizing the internal structure of food substrates. The signal intensity of different sample regions is directly proportional to the water molecule content. Generally, the brighter the color in the image (weighted imaging tends to be red), the stronger the water molecule signal in this region, illustrating the higher water content in a sample. As shown in Figure 4D, the FCST salmon fillets exhibited the smallest red portion, and the green proton density image indicates a lower water content and uneven distribution in the salmon fillet. However, the proton density map of the NNMT sample exhibited a red hue, and water was distributed evenly. The red part in the samples of FMT and NCST lessened, indicating the water content was reduced.

### 3.3. Color

Color is one of the most visual indicators of salmon’s freshness and quality characteristics. As shown in Table 3, the *L** value in FCST was significantly different from the FMT, NCST, and NNMT samples (*p* < 0.05). The increase in *L** value may be due to drip loss (higher content of surface-free water) [35]. The *L** value of the FMT (51.96 ± 0.56) and NNMT (51.99 ± 0.65) treatment groups was higher, which may be caused by the uneven heating of microwave thawing and lead to severe water loss on the surface of salmon. Our experiment stated clearly that the FCST and NCST samples had lower *a** values, indicating that myoglobin was oxidized [36]. The change in the *b** value is influenced by the oxidation of polyunsaturated fatty acids [37]. The *b^*^* value of the FCST sample was the lowest but only decreased by 1.86 compared with the NNMT sample. Therefore, MNPs-assisted freezing–thawing could effectively inhibit protein oxidation from the perspective of color.

### 3.4. TPA and Stress Relaxation Parameters Analysis

Texture characteristics can directly reflect changes in muscle texture and quality. After freezing and thawing, the protein in fish muscle will slowly decompose, the springiness, hardness, and chewiness will decrease to different degrees, and the food quality will also deteriorate. As shown in Table 4, the hardness of FCST, FMT, NCST, and NNMT was significantly different (*p* < 0.05), and the highest hardness was observed in the NNMT group. Qiu et al. [35] believe that reducing the freezable water (free water) content and inhibiting the formation of many ice crystals is beneficial to the sample’s texture during freezing and thawing. NNMT effectively inhibits ice crystal crystallization and recrystallization and protects cell integrity; therefore, NNMT had the highest hardness. The results of chewiness were the same in the FMT, NNMT, FCST, and NCST groups (*p* < 0.05).

The stress relaxation test is an important evaluation tool for determining viscoelastic properties. Relaxation time is an essential parameter in stress relaxation characteristics, and it is the result of the joint action of viscosity and elastic behavior. The stress relaxation time is combined with the binding force between muscle fiber molecules. The stronger the binding force between muscle fiber molecules, the longer the relaxation time required for sliding between them [38]. The relaxation time of the NNMT sample was significantly different from the FCST, FMT, and NCST groups (*p* < 0.05) (Table 4). Meanwhile, the NNMT sample had the longest relaxation time, showing that the interaction between myofibrillar proteins via NNMT treatment was the most obvious, which was more conducive to the quality of salmon fillets and trapped more water molecules.

### 3.5. Analysis of Protein and Lipid Oxidation

#### 3.5.1. Protein Oxidation

Conformational changes occur during the process of protein oxidation, and the external sulfhydryl group is easily oxidized to the disulfide bond, bringing about the decrease of the total sulfhydryl group. Contrasted with FCST, the total sulfhydryl content of MNPs-assisted freezing–thawing samples increased significantly (*p* < 0.05) in Figure 5A. The results showed that the FCST group had the most severe oxidation; MNPs-assisted freezing–thawing could inhibit protein oxidation. Ice crystals cause irreversible damage to cellular structures [39]. Furthermore, ice crystallization and recrystallization may indirectly trigger oxidation [40]. The total sulfhydryl content of the NNMT sample was the highest (88.16 nmol/mg), indicating that NNMT helps rapidly form small ice crystals by changing the free energy barrier of free water nucleation during freezing and further inhibiting recrystallization by microwave rewarming treatment during thawing. It had a positive effect on inhibiting myofibrillar protein oxidation.

Carbonyl groups are generated by the degradation of side chains of amino acid molecules and the cleavage of peptide bonds, leading to the aggregation and cross-linking of myofibrils [41]. In Figure 5A, it can be observed that the carbonyl content of the FCST sample was the highest. The carbonyl content of the NCST sample was significantly different from the FCST sample (*p* < 0.05). This could be due to NNMT’s decreased freezing point and freezing enthalpy, inhibiting the formation of ice crystals and thus delaying protein oxidation. In addition, the carbonyl content of the NNMT sample was the lowest (5.95 nmol/mg). The MNPs in solution under an alternating magnetic field can convert electromagnetic energy into heat. Hence, NNMT could accelerate the thawing rate and achieve an even heating effect in this study.

Dityrosine can reflect the oxidation of amino acids caused by changes in L-tyrosine molecules [41]. The variations in dityrosine content of MP caused by different freezing–thawing treatments are shown in Figure 5B. The relative fluorescence value is utilized to represent the dityrosine content. The NNMT sample exhibited a significantly lower relative fluorescence intensity compared to the FCST, FMT, and NCST samples, indicating that NNMT treatment hindered the interaction between tyrosine free radicals and prevented serious oxidation of MP. The FCST sample had the most vigorous fluorescence intensity as a result of weak bonding interactions between amino acid molecules. MNPs-assisted freezing–thawing can better maintain the structural stability of myofibrillar proteins and thus postpone oxidation.

#### 3.5.2. Lipid Oxidation

The TBA value can reflect the malondialdehyde (MDA) content, which is a secondary lipid oxidation product that generates an unpleasant taste and can be used to judge the degree of oxidative rancidity of lipids in fish meat [42]. A higher TBA value means a more serious degree of lipid oxidation. As a kind of marine fish with fat-rich foods, the unsaturated fatty acids of salmon are easily destroyed by oxidation, thus destroying the muscle cells and leading to the exudation of cell contents. As demonstrated in Figure 5B, the TBA value of the FCST salmon fillet was significantly different from that of FMT and NCST samples (*p* < 0.05), and the NNMT sample was the lowest (0.09 mg MDA/kg). In the process of freezing–thawing, the ice crystals’ formation and recrystallization affected cell integrity and triggered the release of oxidative precursors, thus increasing the extent of lipid oxidation [43]. The TBA value of the NNMT group was the lowest, which could be attributed to the rapid and uniform thawing achieved by nanowarming. In addition, the inferior sensory quality of thawed raw meat and the occurrence of protein oxidation are closely related to lipid oxidation [44].

### 3.6. Analysis of Protein Aggregation and Degradation

#### 3.6.1. Particle Size

Changes in the average particle size of salmon fillets with different freezing–thawing methods are shown in Figure 6A. The disulfide bonds and hydrophobic interactions between protein molecules form polymerization, which increases particle size. In this study, the average particle size of the FCST sample was markedly different from that of FMT, NCST, and NNMT (*p* < 0.05). The average particle sizes of FCST, FMT, NCST, and NNMT were 945.49 nm, 514.57 nm, 507.25 nm, and 461.98 nm, respectively. The NNMT group had the smallest particle size, implying more protein–water interactions and less myofibrillar protein aggregation. Thus, the conformational proteins were stable, maintained the protein–water interaction, and bound more water molecules. This also explained why the NNMT sample exhibited better WHC. The average particle size of NCST was 438.24 nm less than that of FCST, indicating that magnetic nanoparticles could inhibit the aggregation of MP [45]. Zhu et al. found that the particle size of myofibrillar protein is closely associated with protein oxidation and degradation, with severe oxidation being accompanied by an increase in particle size [18].

#### 3.6.2. Solubility of Protein

Solubility is considered an indication of protein function and can shed light on protein denaturation and aggregation. High protein solubility displays better emulsification and water retention. As shown in Figure 6A, the MP solubility of NNMT was the highest (87.28%). It is probable that MNPs increase the thawing rate and inhibit the recrystallization of ice crystals in the process of thawing, resulting in a more complete muscle fiber structure and higher solubility. The MNPs may change the nucleation-free energy barrier and lower the freezing point of the sample, reducing the formation of large ice crystals, which is another reason for the high solubility of the NNMT group. The protein solubility of the FCST sample was significantly different from the samples of FMT, NCST, and NNMT (*p* < 0.05). Previous research from our laboratory has found that protein solubility is inversely proportional to particle size [18].

#### 3.6.3. SDS-PAGE

The SDS-PAGE pattern of MP under different freezing–thawing treatments is shown in Figure 6B. Each lane had more than eight different protein bands. Still, it could be seen that the main bands of salmon myofibrillar proteins include the myosin heavy chain (MHC, about 180 kDa), actin (about 48 kDa), and tropomyosin (about 36 kDa), which accounts for the largest proportion of myofibrillar proteins. When the temperature is below 255 K, changes in surface hydrophobicity, hydrogen bonding, and molecular water density make the original protein structure unstable, which is called the cold denaturation of muscle proteins [46]. The cross-linking and aggregation of proteins lead to the emergence of heavy chains, and the degradation of heavy chains leads to the emergence of light chains [18]. Compared to FCST, FMT, and NCST groups, the MHC band of NNMT was light, and the actin band of NNMT was narrowed. It indicates that the actin cleaved smaller peptides and higher molecular weight polymers were formed by cross-linking non-disulfide [47].

## 4. Conclusions

This study has demonstrated that the incorporation of MNPs during freezing–thawing can enhance the thermodynamic properties of salmon fillets, resulting in a reduction in freezing point and an improvement in *T_g_* and *T_max_*. This process results in the formation of uniformly dispersed, small ice crystals during the freezing process. In the course of thawing, NNMT could restrain the recrystallization of ice crystals and decrease the drip loss by secondary extrusion cells. Moreover, NNMT has the potential to modify textural properties and enhance WHC for better alignment with consumer preferences. NNMT could also ensure the compactness and completeness of salmon muscle fibers, effectively reducing oxidation. In conclusion, NNMT proves to be a viable freezing–thawing technique for enhancing the quality of salmon.

## Figures and Tables

**Figure 1 foods-12-02887-f001:**
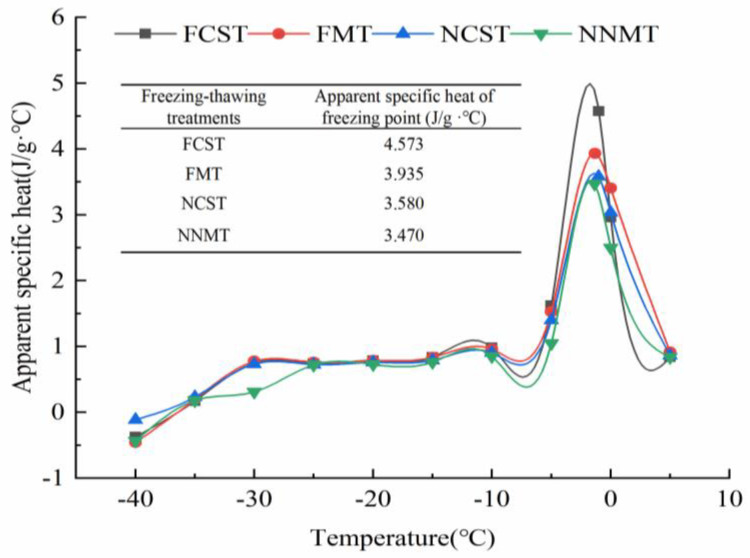
Apparent specific heat of salmon fillets as affected by different freezing–thawing treatment.

**Figure 2 foods-12-02887-f002:**
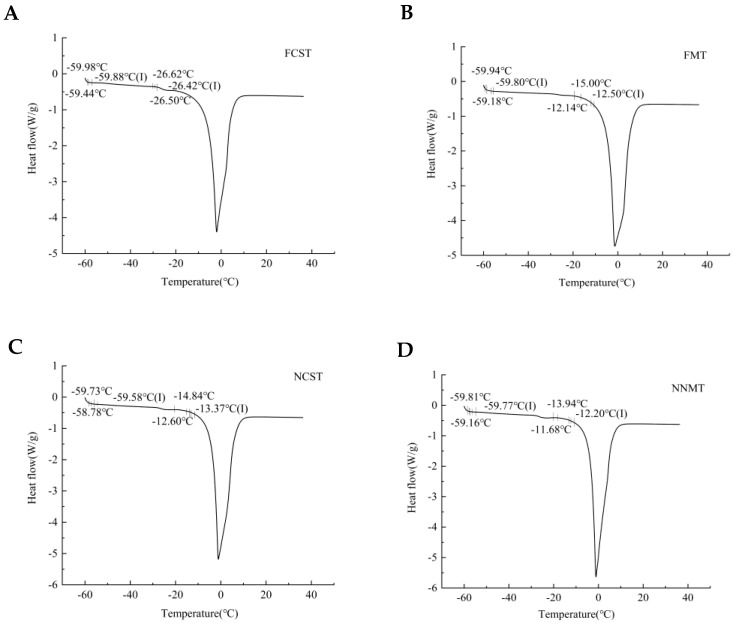
Glass transition temperature of salmon fillets as affected by different freezing–thawing treatment.

**Figure 3 foods-12-02887-f003:**
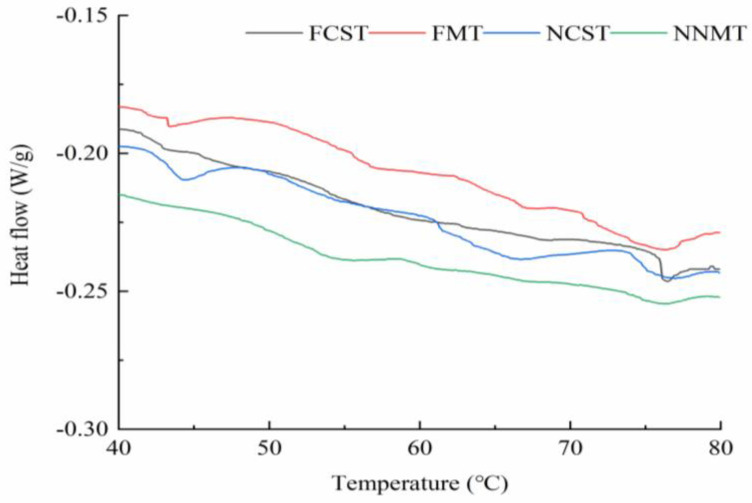
Protein thermal stability of salmon fillets as affected by different freezing–thawing treatment.

**Figure 4 foods-12-02887-f004:**
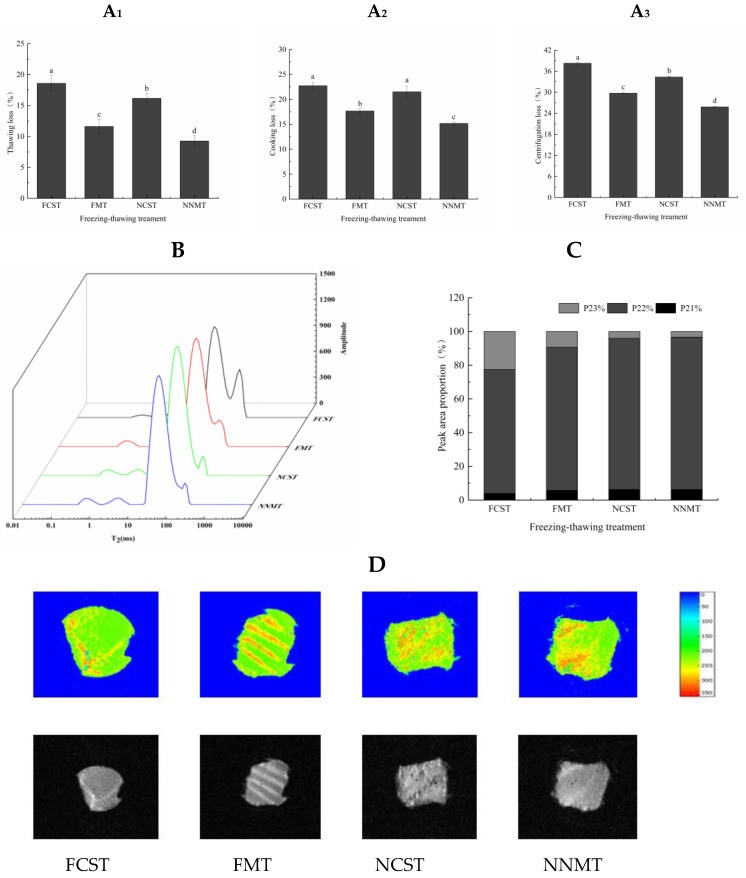
Water retention (**A**), transverse relaxation time *T*_2_ of water (**B**), water distribution and composition, (**C**) and magnetic resonance imaging (**D**) of salmon fillets.

**Figure 5 foods-12-02887-f005:**
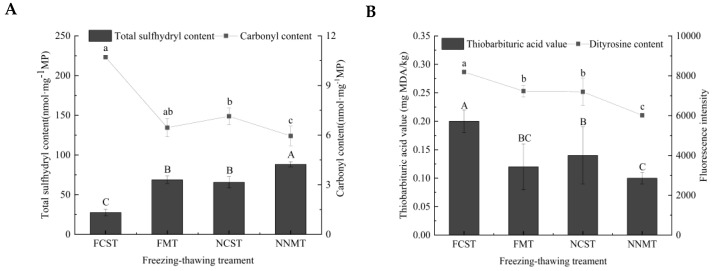
Total sulfhydryl content and carbonyl content (**A**), thiobarbituric acid value (TBA) and dityrosine content (**B**) of salmon fillets.

**Figure 6 foods-12-02887-f006:**
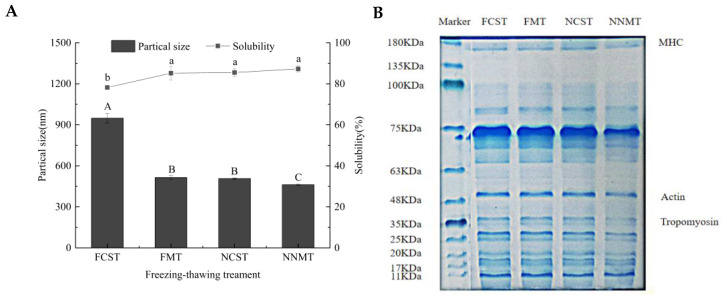
Particle size and protein solubility (**A**); SDS-PAGE (**B**) of salmon fillets.

**Table 1 foods-12-02887-t001:** Effects of freezing–thawing treatment on freezing point and unfreezable water content of salmon fillets.

Treatments	Freezing Point/°C	Moisture Content/%	Freezing Enthalpy/J·g^−1^	Unfreezable Water Content/%
FCST	−1.00 ± 0.02 ^c^	68.11 ± 2.00 ^a^	194.55 ± 1.75 ^a^	12.68 ± 1.42 ^c^
FMT	−1.31 ± 0.01 ^b^	66.98 ± 1.57 ^a^	171.60 ± 8.30 ^b^	18.29 ± 1.00 ^a^
NCST	−1.02 ± 0.02 ^c^	67.81 ± 1.39 ^a^	175.27 ± 2.67 ^b^	15.30 ± 0.84 ^b^
NNMT	−1.35 ± 0.03 ^a^	69.30 ± 1.55 ^a^	161.85 ± 0.95 ^c^	19.31 ± 1.82 ^a^

Different letters (a–c) show statistically significant difference (*p* < 0.05).

**Table 2 foods-12-02887-t002:** *T_max_* and ∆H values obtained by different freezing–thawing treatments for salmon fillets.

Treatments	Peak 1		Peak 2		Peak 3	
	*T_max_*1 (°C)	∆H1 (J/g)	*T_max_*2 (°C)	∆H2 (J/g)	*T_max_*3 (°C)	∆H3 (J/g)
FCST	43.08 ± 0.02 ^c^	0.17 ± 0.01 ^a^	54.06 ± 0.36 ^a^	0.04 ± 0.01 ^b^	74.63 ± 1.50 ^b^	0.06 ± 0.02 ^a^
FMT	42.90 ± 0.43 ^c^	0.18 ± 0.02 ^a^	55.45 ± 1.29 ^a^	0.08 ± 0.01 ^a^	76.14 ± 0.19 ^a^	0.34 ± 0.27 ^a^
NCST	44.09 ± 0.71 ^b^	0.26 ± 0.07 ^a^	56.06 ± 1.31 ^a^	0.05 ± 0.02 ^b^	76.10 ± 1.18 ^a^	0.33 ± 0.21 ^a^
NNMT	44.89 ± 0.16 ^a^	0.13 ± 0.01 ^a^	55.57 ± 0.82 ^a^	0.11 ± 0.02 ^a^	76.58 ± 1.03 ^a^	0.12 ± 0.04 ^a^

Different letters (a–c) show statistically significant difference (*p* < 0.05).

**Table 3 foods-12-02887-t003:** Effects of different freezing–thawing treatment on chromatic aberration of salmon fillets.

Treatments	*L**	*a**	*b**
FCST	49.06 ± 0.27 ^b^	9.64 ± 0.98 ^ab^	16.35 ± 1.46 ^b^
FMT	51.96 ± 0.56 ^a^	10.40 ± 0.41 ^ab^	16.93 ± 0.45 ^ab^
NCST	51.23 ± 0.65 ^a^	9.37 ± 0.50 ^b^	17.04 ± 0.96 ^ab^
NNMT	51.99 ± 0.65 ^a^	10.89 ± 0.52 ^a^	18.21 ± 0.15 ^a^

Different letters (a,b) show statistically significant difference (*p* < 0.05).

**Table 4 foods-12-02887-t004:** Effects of different freezing–thawing treatment on TPA and stress relaxation of salmon fillets.

Treatments	Hardness/g	Springiness/g	Cohesiveness/g	Chewiness/mJ	Relaxation time/s	Relaxation Stress/Pa
FCST	217.40 ± 11.25 ^d^	0.74 ± 0.01 ^a^	0.67 ± 0.03 ^a^	107.42 ± 10.69 ^b^	1.83 ± 0.03 ^b^	1783.10 ± 110.25 ^d^
FMT	523.86 ± 24.91 ^b^	0.75 ± 0.07 ^a^	0.64 ± 0.05 ^ab^	252.02 ± 39.84 ^a^	1.91 ± 0.14 ^b^	2782.45 ± 27.05 ^b^
NCST	271.09 ± 26.05 ^c^	0.74 ± 0.01 ^a^	0.65 ± 0.02 ^ab^	130.58 ± 15.80 ^b^	1.98 ± 0.13 ^b^	2089.75 ± 145.15 ^c^
NNMT	614.19 ± 27.37 ^a^	0.70 ± 0.06 ^a^	0.59 ± 0.04 ^b^	254.33 ± 27.38 ^a^	2.26 ± 0.01 ^a^	3223.98 ± 193.80 ^a^

Different letters (a–d) show statistically significant difference (*p* < 0.05).

## Data Availability

All related data and methods are presented in this paper.

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
