# Peer review of "Nano Freezing–Thawing of Atlantic Salmon Fillets: Impact on Thermodynamic and Quality Characteristics"

_foods, 2023, doi:10.3390/foods12152887_

Round 1

Reviewer 1 Report (Previous Reviewer 1)

Manuscript ID: foods-2473704-peer-review-v1

Nano freezing-thawing of Atlantic salmon fillets: impact on thermodynamic and quality characteristics

I think it's a very interesting and very important topic in seafood context as regard quality assessment. The aim of this study was to estimate the effects of the magnetic nanoparticles assisted cryogenic freezing integrated with MNPs combined microwave thawing (NNMT) on the thermodynamic and quality changes of salmon fillets. The topic is of interest for the academics and for industries; Results are of interest improving the storage stability of salmon fillets. There are few studies like this in literature: this study was original and innovative, well conduced, the research is well performed, the sampling analysis and statistical analysis were well done.

The conclusions are of interest and literature is update.

The manuscript is well written and easy to understand by readers. Revisions are well done and the manuscript suitable.

Author Response

July 23, 2023

Dear Editor:

We are truly grateful to reviewers’ critical comments and thoughtful suggestions about our paper submitted to Foods (Manuscript ID: foods-2511575). Based on these comments and suggestions, we have made careful modifications on the original manuscript. Point-by-point responses to the reviewer and editors’ comments are listed below this letter. All changes made in the revised text using yellow highlighting. We hope that our effort has made this manuscript better and the new manuscript will meet your magazine’s standard. Thank you for providing us the opportunity to revise our manuscript and we look forward to having your editorial decision!

Sincerely yours,

Wenhui Zhu

College of Food Science and Engineering, Bohai University

No. 19, Keji Road Jinzhou 121013, P. R. China

Tel: +86 416 3719190; Fax: +86 416 3719190; E-mail: [email protected]

Foods  

Responses to comments

Title: Nano freezing-thawing of Atlantic salmon fillets: impact on thermodynamic and quality characteristics

Manuscript ID: foods-2511575

Authors: Wenxuan Wang, Wenzheng Li, Ying Bu, Xuepeng Li and Wenhui Zhu*

Responses to reviewers

To Referee 1

I think it's a very interesting and very important topic in seafood context as regard quality assessment. The aim of this study was to estimate the effects of the magnetic nanoparticles assisted cryogenic freezing integrated with MNPs combined microwave thawing (NNMT) on the thermodynamic and quality changes of salmon fillets. The topic is of interest for the academics and for industries; Results are of interest improving the storage stability of salmon fillets. There are few studies like this in literature: this study was original and innovative, well conduced, the research is well performed, the sampling analysis and statistical analysis were well done.

The conclusions are of interest and literature is update.

The manuscript is well written and easy to understand by readers. Revisions are well done and the manuscript suitable.

Response: Thank you for your acknowledge.

The paper has been polished by an expert native English speaking editors with a PhD degree. The references have been adjusted and modified, and other changes in the article have been highlighted, see the revised manuscript.

Reviewer 2 Report (Previous Reviewer 3)

The author had corrected the maniscript according to my comments.

Author Response

July 23, 2023

Dear Editor:

We are truly grateful to reviewers’ critical comments and thoughtful suggestions about our paper submitted to Foods (Manuscript ID: foods-2511575). Based on these comments and suggestions, we have made careful modifications on the original manuscript. Point-by-point responses to the reviewer and editors’ comments are listed below this letter. All changes made in the revised text using yellow highlighting. We hope that our effort has made this manuscript better and the new manuscript will meet your magazine’s standard. Thank you for providing us the opportunity to revise our manuscript and we look forward to having your editorial decision!

Sincerely yours,

Wenhui Zhu

College of Food Science and Engineering, Bohai University

No. 19, Keji Road Jinzhou 121013, P. R. China

Tel: +86 416 3719190; Fax: +86 416 3719190; E-mail: [email protected]

Foods  

Responses to comments

Title: Nano freezing-thawing of Atlantic salmon fillets: impact on thermodynamic and quality characteristics

Manuscript ID: foods-2511575

Authors: Wenxuan Wang, Wenzheng Li, Ying Bu, Xuepeng Li and Wenhui Zhu*

Responses to reviewers

To Referee 2

The author had corrected the maniscript according to my comments.

Response: Thank you for your acknowledge. As for the improvement of the introduction, results and conclusions, we have made corresponding modifications and supplements. See the revised manuscript.

The paper has been polished by an expert native English speaking editors with a PhD degree. The references have been adjusted and modified, and other changes in the article have been highlighted, see the revised manuscript.

Reviewer 3 Report (New Reviewer)

In its present form, it can be further considered if the authors address the following comments and modifications:

The title does not fully reflect the content of the manuscript.

Line 11: “MNPs assisted” is missing a hyphen; please correct it throughout the manuscript.

Line 42: Grammar mistake

Line 80: Grammar mistake

Lines 90-92: What was the microwave thawing time? If it was determined by monitoring the internal temperature of fillets, please specify how the temperature was measured. Same for lines 103-104.

Line 141-142: Briefly explain these methodologies.

Line 179: It is necessary to perform at least two different tests to obtain reliable results on lipid oxidation since a single experiment (TBA method in this case) does not provide complete information regarding lipid deterioration over time.

Line 187-188: This methodology is not clear enough.

Lines 218-220: The discussion or explanation of how nanoparticles on treatment NNMT reduced the freezing point is weak; the phrase “promoting heterogeneous nucleation” is quite simple.

Line 225: It is stated in Figure 1 that no significant were detected between -40 and -10°C among the treatments, and it rose significantly between -10 and 5°C; however, no SD bars are included within the graph.

Lines 377-384: Please explain how the authors infer that low a* values result from myoglobin oxidation and that b* changes are related to PUFA oxidation. Moreover, it would be desirable to include discussion regarding the results of L* values.

Line 405: The name of the table is wrong.

Lines 387-404: The results of the TPA are merely descriptive, and no discussion or explanation of the obtained results is included. How did the different thawing methodologies affect the texture? Did the presence of nanoparticles influence the TPA results?

Line 453: What oxidative precursors were released after ice crystal formation and recrystallization?

Minor mistakes were detected. 

Author Response

July 23, 2023

Dear Editor:

We are truly grateful to reviewers’ critical comments and thoughtful suggestions about our paper submitted to Foods (Manuscript ID: foods-2511575). Based on these comments and suggestions, we have made careful modifications on the original manuscript. Point-by-point responses to the reviewer and editors’ comments are listed below this letter. All changes made in the revised text using yellow highlighting. We hope that our effort has made this manuscript better and the new manuscript will meet your magazine’s standard. Thank you for providing us the opportunity to revise our manuscript and we look forward to having your editorial decision!

Sincerely yours,

Wenhui Zhu

College of Food Science and Engineering, Bohai University

No. 19, Keji Road Jinzhou 121013, P. R. China

Tel: +86 416 3719190; Fax: +86 416 3719190; E-mail: [email protected]

Foods  

Responses to comments

Title: Nano freezing-thawing of Atlantic salmon fillets: impact on thermodynamic and quality characteristics

Manuscript ID: foods-2511575

Authors: Wenxuan Wang, Wenzheng Li, Ying Bu, Xuepeng Li and Wenhui Zhu*

Responses to reviewers

To Referee3

  1. Comment: The title does not fully reflect the content of the manuscript.

Response: Thank you for your suggestion. To accentuate the fundamental content of this investigation, the title underscores pivotal details such as nano freezing-thawing, thermodynamic characteristics, and quality.

  1. Comment: Line 11: MNPs assisted is missing a hyphen; please correct it throughout the manuscript.

Response: Thank you for your suggestion. We have made corresponding modifications. See line 11.

  1. Comment: Line 42: Grammar mistake

Response: Thank you for your suggestion. We have made corresponding modifications. See line 42.

  1. Comment: Line 80: Grammar mistake

Response: Thank you for your suggestion. We have made corresponding modifications. See line 80.

  1. Comment: Lines 90-92: What was the microwave thawing time? If it was determined by monitoring the internal temperature of fillets, please specify how the temperature was measured. Same for lines 103-104.

Response: Thank you for your suggestion. In this study, the temperatures of the salmon fillets were monitored in real time by a temperature recorder (Applent Precision Instrument Co., Ltd., Changzhou, China), linking a K-type thermocouple. We have made corresponding modifications. See the revised manuscript.

  1. Comment: Line 141-142: Briefly explain these methodologies.

Response: Thank you for your suggestion. We have made corresponding modifications. See the revised manuscript.

  1. Comment: Line 179: It is necessary to perform at least two different tests to obtain reliable results on lipid oxidation since a single experiment (TBA method in this case) does not provide complete information regarding lipid deterioration over time.

Response: Thank you for your suggestion. The TBA value is an essential index of lipid oxidation rancidity. After consulting numerous sources, we discovered that a significant portion of the literature solely employs TBA as an indicator for lipid oxidation.

   Effect of high voltage electrostatic field thawing on the functional and physicochemical properties of myofibrillar proteins, Innovative Food Science & Emerging Technologies, DOI: 10.1016/j.ifset.2019.102191;

  Evaluating the effects of nanoparticles combined ultrasonic-microwave thawing on water holding capacity, oxidation, and protein conformation in jumbo squid (Dosidicus gigas) mantlesFood chemistry, DOI:10.1016/J.FOODCHEM.2022.1342500;

  Effects of nanowarming on water holding capacity, oxidation and protein conformation changes in jumbo squid (Dosidicus gigas) mantles, LWT - Food Science and Technology, DOI:10.1016/j.lwt.2020.109511.

  1. Comment: Line 187-188: This methodology is not clear enough.

Response: Thank you for your question. We have made corresponding modification. See the revised manuscript.

  1. Comment: Lines 218-220: The discussion or explanation of how nanoparticles on treatment NNMT reduced the freezing point is weak; the phrase promoting heterogeneous nucleation is quite simple.

Response: Thank you for your question. We have made corresponding modifications and supplements. See line 226-230.

  1. Comment: Line 225: It is stated in Figure 1 that no significant were detected between -40 and -10°C among the treatments, and it rose significantly between -10 and 5°C; however, no SD bars are included within the graph.

Response: Thank you for your suggestion. After consulting various literature, we have found that it is generally not recommended to include SD bars in the graph of apparent specific heat.

(Tavman, S., Kumcuoglu, S., & Gaukel, V. (2007). Apparent specific heat capacity of chilled and frozen meat products. International Journal of Food Properties, 10(1), 103-112. DOI: 10.1080/ 10942910600755151)

  1. Comment: Lines 377-384: Please explain how the authors infer that low a* values result from myoglobin oxidation and that b* changes are related to PUFA oxidation. Moreover, it would be desirable to include discussion regarding the results of L* values.

Response: Thank you for your question. We have made corresponding modifications. The a* value is caused by the oxidation of myoglobin, this conclusion appears in the reference (Influence of high pressure on the color and microbial quality of beef meat, LWT - Food Science and Technology, DOI:10.1016/S0023-6438(03)00082-3). The b* value changes are related to PUFA oxidation, which appeared in reference (Improving the quality and safety of frozen muscle foods by emerging freezing technologies: A review, Critical reviews in food science and nutrition, DOI: 10.1080/10408398.2017.1345854). For the L* value discussion, we have made corresponding modifications. See the revised manuscript.

  1. Comment: Line 405: The name of the table is wrong.

Response: Thank you for your suggestion. We have made corresponding modifications. See the revised manuscript.

  1. Comment: Lines 387-404: The results of the TPA are merely descriptive, and no discussion or explanation of the obtained results is included. How did the different thawing methodologies affect the texture? Did the presence of nanoparticles influence the TPA results?

Response: Thank you for your question. We have made corresponding modifications. See the revised manuscript. The potential impact of nanoparticles on the outcomes of TPA necessitates further investigation.

  1. Comment: Line 453: What oxidative precursors were released after ice crystal formation and recrystallization?

Response: Thank you for your question. We have made corresponding modifications. See line 472.

The presence of myoglobin enables the absorption of free radicals, leading to the oxidation of OxyMb into MetMb. Consequently, the resulting oxidized ferric ions play a crucial role in promoting lipid oxidation. Upon formation of MetMb from OxyMb, superoxide anions are generated and subsequently combine with hydrogen ions to produce hydrogen peroxide. This process further facilitates the oxidation of both OxyMb and unsaturated fatty acids, thereby enhancing lipid oxidation (Effectiveness of different myoglobin states to minimize high pressure induced discoloration in raw ground beef. Lwt doi: 10.1016/j.lwt.2018.03.008; Interrelationship among ferrous myoglobin, lipid and protein oxidations in rabbit meat during refrigerated and superchilled storage. Meat Science doi: 10.1016/j.meatsci.2018.08.006.). Hence, the oxidative precursor released is believed to be the myoglobin oxidation products.

Thanks again for your valuable suggestions on this study.

The paper has been polished by an expert native English speaking editors with a PhD degree. The references have been adjusted and modified, and other changes in the article have been highlighted, see the revised manuscript.

This manuscript is a resubmission of an earlier submission. The following is a list of the peer review reports and author responses from that submission.

Round 1

Reviewer 1 Report

Manuscript ID: foods-2473704-peer-review-v1 

Nano freezing-thawing of Atlantic salmon fillets: impact on thermodynamic and quality characteristics 

I think it's a very interesting and very important topic in seafood context as regard quality assessment. The aim of this study was to estimate the effects of the magnetic nanoparticles assisted cryogenic freezing integrated with MNPs combined microwave thawing (NNMT) on the thermodynamic and quality changes of salmon fillets. The topic is of interest for the academics and for industries; Results are of interest improving the storage stability of salmon fillets. There are few studies like this in literature: this study was original and innovative, well conduced, the research is well performed, the sampling analysis and statistical analysis were well done.  

The conclusions are of interest and literature is update. 

The manuscript is well written and easy to understand by readers. I believe that this manuscript does not need big changes.

Reviewer 2 Report

0) I am sure that the introduction of Fe3O4 MNPs with sizes of 20–50 nm into a food product cannot be safe. Inorganic nanoparticles are banned from use in food in most jurisdictions. A lot of works have shown their passage through most barriers in the human and animal body and the manifestation of dangerous pathological effects - mutagenic, oxidative, carcinogenic, etc.

1) The authors have already published several works with completely similar methodology, but different objects [7,13,14] . This study is largely secondary. It would be logical to explain the expected difference between the previously studied object of study and the object of this article.

2) It is not indicated how many nanoparticles and to what depth penetrated the salmon fillet. Accordingly, what mass/quantity of nanoparticles will a person use together with 100 g of such products?

3) If the distribution of nanoparticles is uneven over the depth of the fillets (not shown), then cutting out a small piece from a large piece (point 2.3) is incorrect from the point of view of the adequacy of the results obtained.

4) Excessive precision in the indication of some standard deviations in the tables and, accordingly, in the average values.

5) Figure 1 lacks an indication of a statistical significance estimate.

6) In Figure 4 A1, melting losses are too high.

7) An unexplained phenomenon of an increase in the amount of non-freezed water with an increase in the glass transition temperature is observed.

8) Taking not the initial, but freezing-thawing treated samples for the study of thermophysical characteristics raises great doubts about the quality of the study and the results obtained.

Reviewer 3 Report

This is an interesting article for preserving salmon fillets, all figures are perfect, but I still have some comments for this article:

1.Why using the title of nano freezing thawing?

2.The author did not describe the method to preparing the magnetic nanoparticle.

3.There are some problems of the format of this article?

4.How about the toxicity of MNPs?

5.In the previous study, the author had used this technoques for preserving jumbo squid, see bass, and others, what are the differences among these samples?

6.The author should add more recent studies about the MNPS in the introduction section.

7.Some method describe too simple (2.4., and 2.7.)

8.Why the treatment of NNMT can increase Tg and enhance the storage stability of salmon fillets?

9.In line 425, what is nanowarming? why it can attribute rapid thawing, and the TBA value was low?

Extensive editing of English language required